# Development and nutritional evaluation of pomegranate peel enriched bars

Rameeza Abbas[1], Muhammad Aamir[1], Farhan Saeed[2], Amar Shankar[3], Jaspreet Kaur[4], Rutaba Nadeem[5], Ashish Singh Chauhan[6], Ali Imran[1], Muhammad Afzaal[1]*, Abdela Befa Kinki[7]*

1 Natiional Institute of Food Science and Technology, University of Agriculture, Faisalabad, Pakistan, 2 Department of Food Sciences, Government College University, Faisalabad, Pakistan, 3 Department of Food Technology, School of Engineering and Technology, JAIN (Deemed to be University), Bangalore, Karnataka, India, 4 Department of Nutrition and Dietetics, Chandigarh Group of Colleges, Jhanjeri, Mohali, Punjab, India, 5 Department of Clinical Nutrition, NUR International University, Lahore, Pakistan, 6 Division of Research and Innovation, Uttaranchal Institute of Pharmaceutical Sciences, Uttaranchal University, Dehradun, India, 7 Department of Food Science and Nutrition, Ethiopian Institute of Agricultural Research, Shashemene, Ethiopia

* muhammadafzaal@gcuf.edu.pk (MA); befabdela@gmail.com (ABK)

**Data Availability Statement:** All relevant data are within the manuscript and its Supporting Information files

**Funding:** The author(s) received no specific funding for this work.

## Abstract

Pomegranate peel powder is used as a functional ingredient in the development of nutritional bars. Pomegranate (*Punica granatum*) is well known fruit belongs to punicaceae family having multiple health benefits, not only limited to its edible parts but also in its non-edible parts mostly the peel. Fruit wastes are rich source of nutrients, and can be used for the development of functional food products. Pomegranate peel is considered to be beneficial due to its functional and therapeutic properties as it is a source of many biological active components like polyphenols, tannins and flavonoids. Nutrient rich and ready-made foods are the demand of everyone due to their easy availability and cost effectiveness. Among the confectionary products, bars are liked by individuals of different age groups. Hence, nutritional properties of bars can be enhanced by using pomegranate peel powder. The current study was designed to develop bars enriched with pomegranate peel powder as a basic ingredient. Pomegranate peel powder is prepared and analyzed for proximate, mineral, total phenolic content, total flavonoid content and anti-oxidant potential (DPPH). By using pomegranate peel powder, oats and jaggery, bars were prepared. In this research, five treatments $T_0$ (0% pomegranate peel powder and 100% oats). $T_1$ (5% pomegranate peel powder and 95% oats), $T_2$ (10% pomegranate peel powder and 90% oats), $T_3$ (15% pomegranate peel powder and 85% oats) and $T_4$ (20% pomegranate peel powder and 80% oats) were used. The developed product is analyzed for proximate, mineral, total flavonoid contents, total phenolic content and anti-oxidant potential (DPPH). Proximate analysis of bars revealed that moisture, protein, fat, fiber, ash and nitrogen free extract ranges from $T_0$ to T4 (13.38 ±1.21 to 11.32±1.15, 9.56±0.92 to 8.32±1.14, 9.05±1.21 to 7.93±1.08, 5.23±0.82 to 16.89 ±0.64, 2.05±0.87 to 2.92±1.25 and 62.51±0.85 to 52.62±0.93 respectively. Phytochemical analysis of bars enriched with pomegranate peel powder revealed that total phenolic content, total flavonoid content and antioxidant potential of bars ranges from $T_0$ to $T_4$ (142.74 ±0.65 to 211.79±0.63 mg GAE/100g, 129.16±0.64 to 192±0.53 mg QE/100g and 41.35

**Competing interests:** The authors have declared that no competing interests exist.

±0.82 to 64.57±0.69%) respectively. Mineral analysis of bars enriched with pomegranate peel powder revealed that calcium, Phosphorus, Potassium, Iron, Magnesium content ranged from $T_0$ to $T_4$ (25.42±0.63 to 31.06±0.58, 51.00±1.01 to 45.05±1.09, 59.46±1.13 to 79.15±0.28, 1.32±1.20 to 1.95±0.83 and 54.17±0.88±0.58 to 57.36±0.68 mg/100g respectively). Sensory evaluation is done for color, aroma, taste, texture overall acceptability. $T_3$ got maximum score. Then, the data obtained were evaluated by CRD design. On the basis of results revealed that treatment $T_3$ with 15% pomegranate peel powder was overall highly acceptable.

## Introduction

Due to changing dietary patterns and over population in the world, the demand of food products has been increased significantly. Fruits and vegetables are essential in human diets. Yet, nearly one-third of food produced for human consumption, about 1.3 billion tons valued at around 990 billion dollars, is lost or wasted during processing. The vegetable and fruit processing industry is responsible for about 45% of these by-products, leading to significant resource wastage and environmental pollution if not managed properly [1]. These by-products, are rich in bioactive compounds such as phenolics, vitamins, minerals, anti-oxidants, and fibers, and can be repurposed as functional foods to combat malnutrition and health issues [2].

In the European Union, 89 million tons of food waste are generated annually, a figure expected to rise significantly. In India, approximately 40% of food produced is wasted, with fruit and vegetable losses estimated at 12 and 21 million tons respectively, resulting in a total food value loss of 10.6 billion USD (FAO). Fruit and vegetable waste (FVW), including inedible components discarded during collection, handling, and processing, contains high levels of phytochemicals, making it valuable for the food, pharmaceutical, textile, and cosmetics industries [3].

Pomegranate, known for its health benefits, is predominantly produced in Iran, where it is also used as herbal medicine. The fruit's peel, constituting 40–50% of its weight, contains more active constituents than the edible part, providing significant health benefits such as anti-oxidant, antibacterial, anti-inflammatory, and anti-cancer properties [3–6]. Pomegranate peel is a rich source of phenolic compounds and has been used in traditional medicine for treating various diseases. Usually discarded as waste product. So, utilizing peel powder is a sustainable approach [7–12].

Oats (*Avena sativa L.*), a nutrient-rich cereal grain, are gaining popularity due to their health benefits, including cholesterol reduction. They are a rich source of proteins, fibers, vitamins, minerals, and bioactive compounds such as polyphenols. Oats are consumed in various forms, from whole grains to bran, offering anti-oxidant, anti-inflammatory, and cholesterol-lowering effects. Despite their benefits, oats account for only 0.86% of global cereal production, although they are a staple in many countries for both human and livestock nutrition [13–16]. Oats are unique for their globulin protein content, unlike most cereals high in prolamins. They contain the most fat among cereals, with low saturated fats and high essential unsaturated fatty acids that reduce cardiovascular disease risk. Oats are a rich source of antioxidants known as avenanthramides that express strong antioxidative activity; because of these benefits, oat-based foods are increasingly becoming popular as functional foods [17]. Oats containing beta-glucan which reduce LDL-Cholestrol by 12.2% after 4 weeks supplementation and 15.1% after 8 weeks. Total cholesterol of patients decreased by 6.5% [18].

The lifestyle changes have increased the consumption of snacks and fast food. Therefore the demands for its production have been increased. Snacks, particularly those enriched with fibers and anti-oxidants from fruits, are seen as healthy and convenient. The demand for nutritious, convenient foods has led to a rise in snack bars, with the market expected to grow from 15 billion USD in 2019 to 19 billion USD by 2025. These bars cater to the nutritional needs of various consumers, especially active individuals like athletes [19, 20].

Thus, the effective utilization of food by-products, particularly from fruits and vegetables, can address environmental issues while promoting health through enriched functional foods. The study focuses on the use of pomegranate peel as a functional ingredient in food products-a typically discarded by-product from fruit waste, therefore adding value to its sustainable development into food products. Emphasizing the nutritional benefits of foods like pomegranate and oats, along with the growing popularity of nutritious snack options, reflects a broader trend towards sustainable and health-conscious consumption in response to global food challenges. The objectives of this study include developing functional bars, exploring the effect of pomegranate peel enrichment, and conducting physicochemical and nutritional analysis of the product.

## Materials and methods

### Procurement of raw materials and treatment

All the research related work was performed in postgraduate research laboratories of National Institute of Food Science and Technology (NIFSAT), University of Agriculture Faisalabad. For the preparation of bars, pomegranate peel, jaggery and rolled oats were required. These raw materials will be obtained from Metro Faisalabad, Pakistan and pomegranate peel (Qandhari variety) from food industry of Faisalabad, Pakistan. Firstly, pomegranate peel was washed to remove dirt, dust and impurities. Then peel was dried at room temperature (25–40°C) and grind it by using the electrical grinder. Pomegranate peel and oats were weighed as in the treatment plan (Table 1) and store in polythene bags for product development and further analysis. Bars enriched with pomegranate peel powder was developed using roasted rolled oats and jaggery. Jaggery was caramelized and the oats were roasted (Figs 1–4). The pomegranate peel powder was added in oats in different combinations according to the treatment plan as mentioned in Table 1 to prepare bars while the amount of jaggery was remain constant. Then mixture was poured in the oil greased container, cut into bars and stored in airtight container at room temperature.

### Proximate analysis of pomegranate peel powder

Raw material (pomegranate peel) was analyzed to determine its nutritional caharacteristics including proximate analysis, phytochemical and mineral analysis. Proximate analysis of dried pomegranate peel powder was done according to methods described in AOAC [21].

**Table 1. Treatments plan used for the preparations of pomegranate peel enriched bars.**

| Treatments | $T_0$ | $T_1$ | $T_2$ | $T_3$ | $T_4$ |
|---|---|---|---|---|---|
| Oats % | 100 | 95 | 90 | 85 | 80 |
| Pomegranate peel powder % | 0 | 5 | 10 | 15 | 20 |

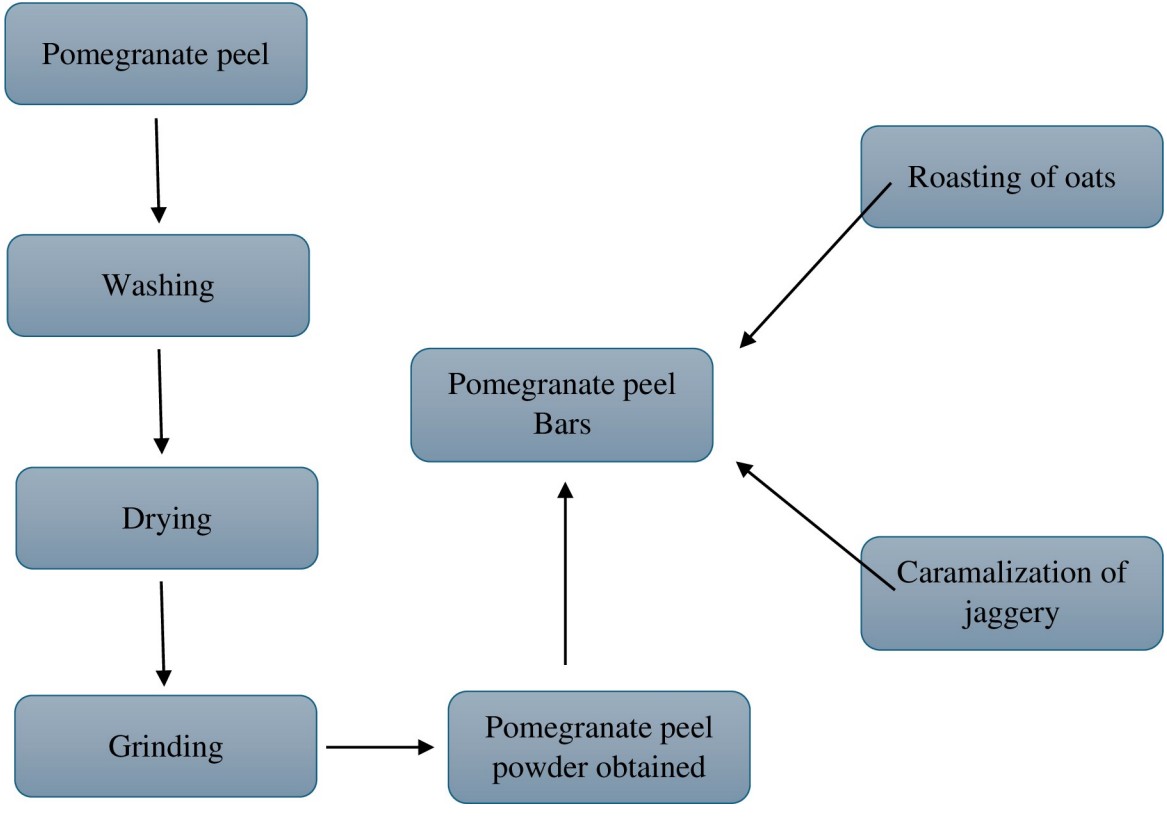

**Fig 1. Flow diagram of preparation of pomegranate peel enriched bars.**

## Moisture

Moisture content of pomegranate peel powder was determined by the standard methods of AOAC [21]. Method no. 934–01. The sample material was taken in the pre-weighed china dishes and kept in hot air oven (Memmert, Germany) at the temperature of 100–110˚C until it attains constant weight. After drying in the dehydrator the sample was removed fro dehydrator and kept in the vaccume desicator. The moisture of the sample was determined by using following formula:

$$Moisture\ (\%) = \frac{weight\ of\ original\ sample\ (g) - weight\ of\ dried\ sample}{weight\ of\ original\ sample\ (g)} \times 100$$

## Crude protein

Protein content in the raw material was determined by the Kjheldal apparatus (Model: D-40599, Behr Labor Technik, Germany) according to the protocols given by AOAC [21] Method No. 984-13Reagents used in this procedure were sulfuric acid (95–98%), digestion mixture (100 g $K_2SO_4$, Cu $SO_4$, 5 gFe $SO_4$) 40% of sodium hydroxide. 0.1 N sulfuric acd for titration, boric acid solution (4%) and 3 drops of methyle red indicator. There are three steps of this procedure disgestion, distilation and titration. 2 g of the sample was weighed and transferred into the digestion flask then 5 g of the digestion tablet and 30 ml concentrated sulfuric acid were added in the flask. The sample was then digested for two to three hours. When the

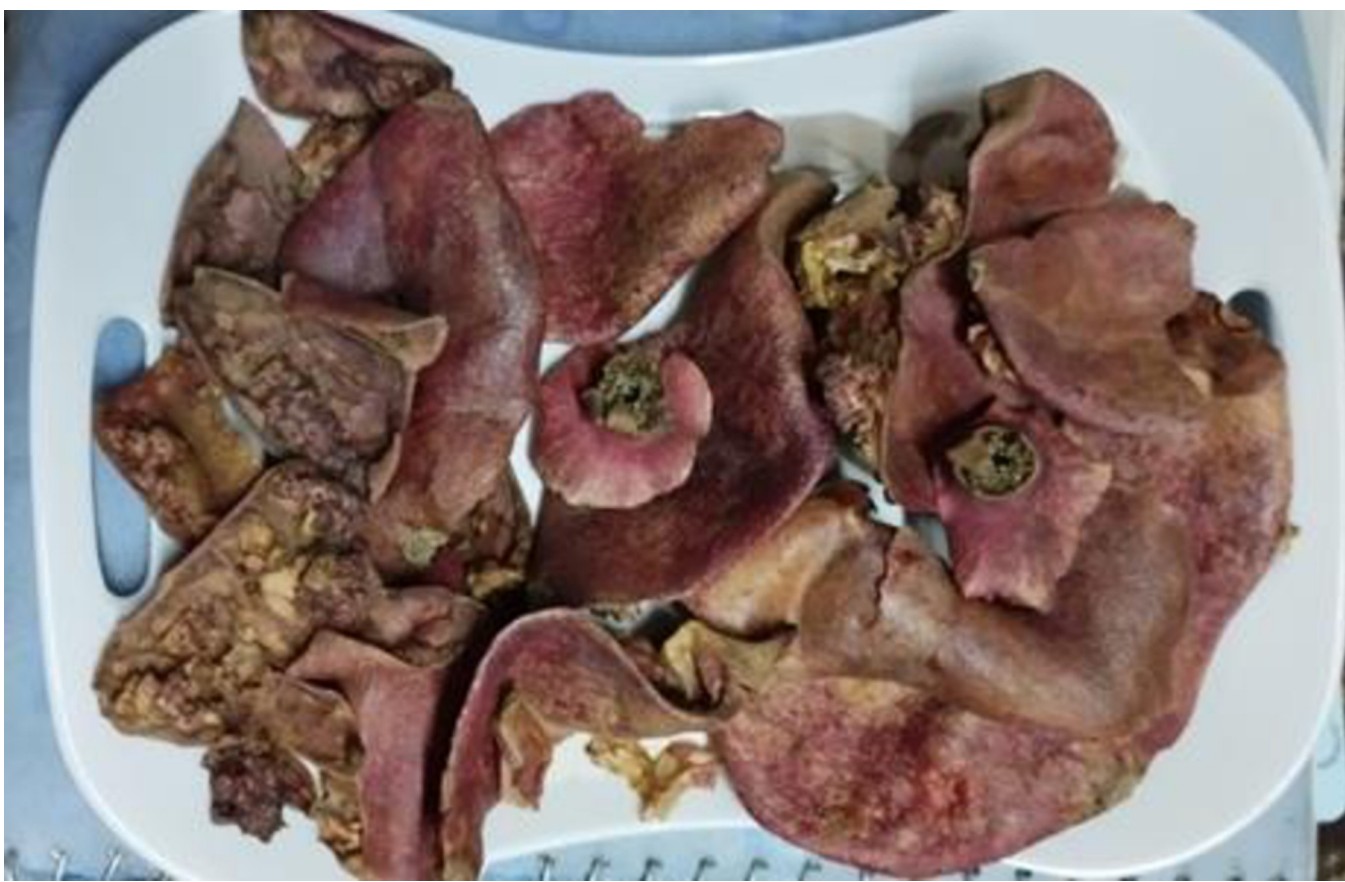

**Fig 2. Pomegranate peel.**

bright green dye was appeared, mixture was cooled for some time. Then sample was poured in to flask of 250 ml and distilled water was added to create volume was taken into kjeldal diges-tion tube and 10 ml sodium hydroxide (40%) was added in it. After that 10.0 ml of boric acid

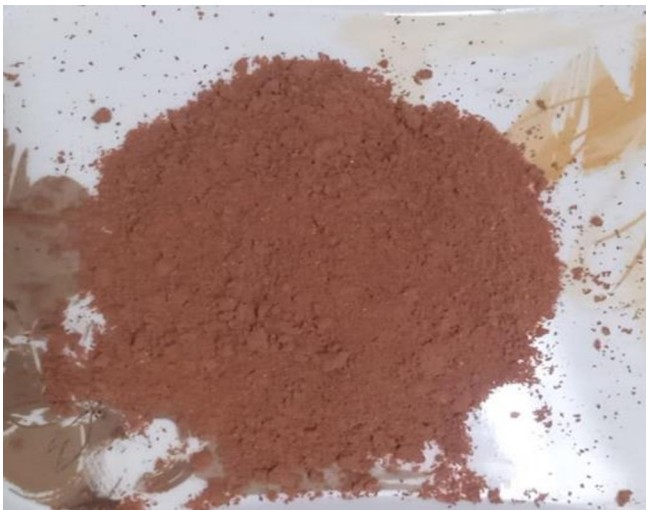

**Fig 3. Pomegranate peel powder.**

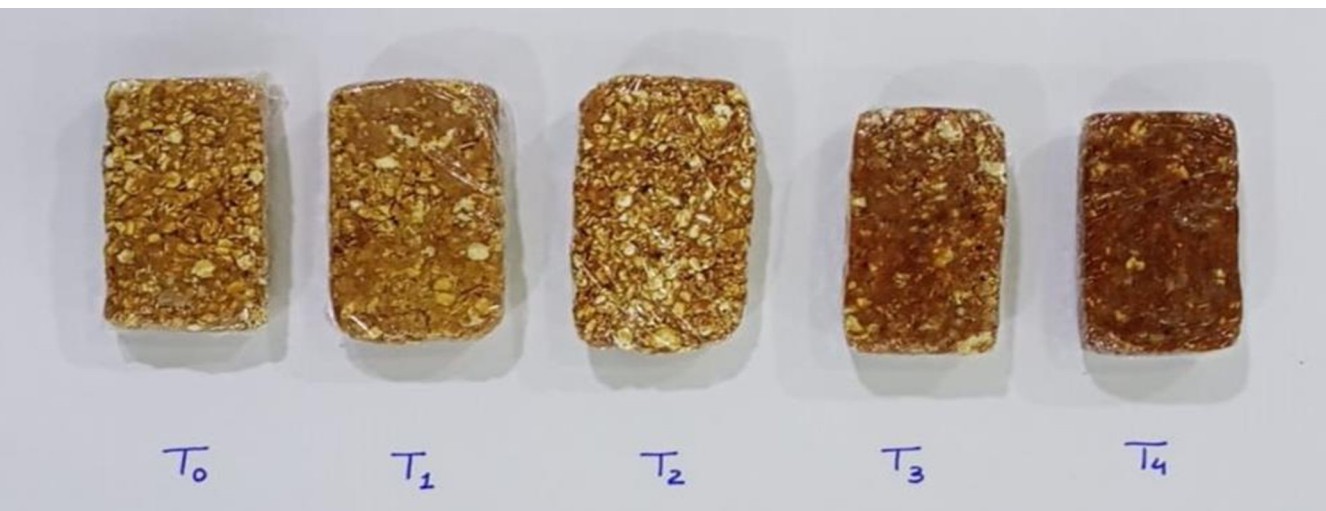

**Fig 4. Bars enriched with pomegranate peel powder.**

(4%) and drops of methyle red indicator (2–3 drops) were added in the distillate and this process was continued until the yelow-red color of boric acid was achieved. Later on, the sample wass titrated with 0.1 N sulfuric acid to achieve pink color as end point. Nitrogen percentage was calculated by using the following formula:

$$\text{Nitrogen (\%)} = \frac{\text{Vol.o H2SO4 used} \times \text{Dilution vol. (250mL)}}{\text{initial weigt of sample (g)} \times \text{Vol.of diluted sample taken}} \times 100$$

In the sample crude protein % was calculated by multiplying percentage of nitrogen with 6.25 factor

$$\text{Crude protein (\%)} = \text{Nitrogen (\%)} \times 6.25$$

## Crude fat

Crude fat was measured with the help of Soxhlet apparatus (Model: H-2 1045 Extraction Unit, Hoganas, Sweden) by using diethyl ether mentioned in AOAC [21] Method no. 920–39. 5 g samples (moisture free) were taken in thimbles made of whatsman filterpaper and then these thimbles were put in the extraction portion of Soxhlet apparatus. Samples were kept in extraction thimble of soxhlet apparatus covered with plug of cotton and have a cold water arrangement. Solvent used in the process was n-hexane. The heating temperature was set in a way that hexane droplets fell in a continuous manner at trail kept in abstraction portion. Samples were subjected for fat extraction with hexane (50mL) by adjusting the rate of 3–4 drops of hexane per second. After 6–7 siphons thimble was removed and placed in an oven at 105˚C for 3–4 hours to evaporate the solvent. Following formula was used for calculation.

$$\text{Crude fat (\%)} = \frac{\text{initial weight of sample (g)} - \text{final weight of sample}}{\text{initial weight of sample (g)}} \times 100$$

## Crude fiber

Dietary fiber content of the raw material was determind by the Fibertech (VELP Scientifica FIWE Raw Fiber Extractor, Italy) given by AOAC [21]. Fat extracted samples of pomegranate peel wes evaluated to determine crude fiber according to Method no. 32–10.01, as described in AACC [21] and Method no. 978–10, as described in AOAC [21]. 2 g of samples (moisture and fat free) were heated in beakers for half hour for digestion. Afterwards they were filtered with Whatman filter paper along with filtration flask and washed thrice with hot water. The residue were then moved into another 500 ml capacity beakers with a 200 ml of 1.25% NaOH solution. The exact procedure was repeated again until samples become alkali free. They were then transferred to a china dish and put into oven for drying at 100°C for duration of 3 to 4 hours until constant weight was achieved. The samples were then heated on flame until smoke free and kept into the muffle furnace at 550°C for 4 hours until the ash was obtained. The samples were then cooled in a desiccator and weighed. The following obtained data was recorded appropriately and formula was used for the calculation:

$$\text{Crude fiber (\%)} = \frac{\text{weight of dried sample after digestion (g)} - \text{weight of ash}}{\text{initial weight of dried and defatted sample (g)}} \times 100$$

## Ash

Pomegranate peel powder was analyzed to detemine ash content in the sample according to the method explained by AOAC [21] Method no. 942–05. 3 g of the sample wes taken in the crucible and placed on diret flame of burner for charring until no fumes were left in the sample. Then sample was incinerated in the muffle furnance at 550–600°C temperature for 5–6 hours until the sample is turned greyish white color residue. Ash content of the sample was determned by using the following formula:

$$\text{Ash (\%)} = \frac{\text{weight of residues after incineration (g)}}{\text{initial weight of sample}} \times 100$$

## Nitrogen free extract

It was determined by subtracting the percentages of moisture, crude protein, crude fiber, fat and ash from hundred

$$\text{NFE (\%)} = 100 - (\% \text{ moisture} + \% \text{ crude fat} + \% \text{ crude protein} + \% \text{ ash} + \% \text{ crude fiber})$$

## Phytochemical analysis of pomegranate peel powder

**Total phenolic content (TPC).** To determine TPC of pomegranate peel, sample was prepared by taking 1 g of pomegranate peel powder in flask and 10 ml of methanol solution (80%) was added in flask. After this flask was kept for 15 minutes in ultrasonic bath. The extract was centrifuged for 5 minutes at speed of 5000 rpm. After centrifugation, it was filtered with Millipore membrane filter. For determination of TPC, 200uL of aliquot of pomegranate peel was mixed with 100 uL of Folin-Ciocalteu regent and 800 uL of deionized water. After this addition of 300 uL of 20% w/v sodium carbonate was poured in developed solution was done and incubated in dark for 2 hours at 25°C. The absorbance was measured at 765 nm with UV/via

spectrometer (PG instruments, T80). Calibration curve of Gallic acid standard was achieved. TPC was represented as GA equivalent/g [22].

$$C = c \times \frac{V}{m}$$

C = total contents of phenoliccompounds in mg/g
c = gallic acid concentration
V = extract volume
m = Extract weight in grams

**Total flavonoid content (TFC).**   For the analysis of TFC, aliquot of 2% $AlCl_3$ which was 0.5mL was prepared in ethanol solution and added in 0.5 Ml of sample solution. After this these were incubated for 1 hour at room temperature. Afterwards absorbance was measured at 420 nm in spectrophotometer (PG instruments, T80). TFC was calculated from calibration curve as quercetin equivalent (mg/g) [23].

## Antioxidant analysis

Antioxidant test measures DPPH which was measured by respective method of [24]. In order to analyse the scavenging activity of phenolic compounds, 2, 2-diphenyl-1-picry lhydrazyl (DPPH) reagent was used. DPPH reagent prepared by dissolving 60M DPPH in 2mL ethanol. The 0.5ml of pomegranate peel powder was added. Blank sample was prepared using 0.5ml of 99% ethanol without any extract. The observance measured at 517m by using UV-VIS, spectrophotometer (PG instruments, T80). Then the calculation of inhibition rate of DPPH free radical (IR) was done.

$$IR = \frac{A0 - As}{A0} \times 100$$

In the formula;
A0 = blank absorbance
As = sample absorbance of sample

## Mineral analysis of pomegranate peel powder

The pomegranate peel was analyzed for the presence of minerals according to the method described in AOAC [21]. The mineral analysis was done for biotin and potassium after the wet digestion through atomic absorption spectrophotometer. The wet digested sample was atomized. It absorbed energy in ground state and moved to excited state and the energy of a specific wavelength was absorbed from a hollow cathode lamp. The difference between the amount of energy absorbed and emitted was measured. The absorption was linearly related to concentration. The quantity of 3–5 g flour was taken into a conical flask. Then HCL and $HNO_3$ were added in it at a ratio of 7:3 and placed the flask on a hot plate. The sample was heated with acids for 3 to 4 hours until a transparent and around 1–2 ml of the residue was left in flask. After that, the flask was removed from the hot plate, cooled and the volume was made with distilled water up to 100–250 ml. standard curve was obtained by running samples of known strength for each mineral. Standard curve prepared for each sample was used to determine the mineral content of the samples.

## Analysis of pomegranate peel powder bars

**Proximate analysis.**   Crude protein, crude fat, Ash content, crude fiber, moisture content and nitrogen free extract analysis of the prepared product was done by using respective

method of AOAC [21]. Minerals analysis of the product was performed by following procedure of AOAC [21].

**Total phenolic contents.** TPC was determined by using Folin-Ciocalteau reagent according to the method described by [22].

**Total flavonoid contents.** TFC was determined according to the method described by [23].

**Antioxidant activity DPPH.** DPPH was determined using the protocol described by [24].

## Sensory evolution

Pomegranate peel enriched bars were evaluated for sensory characteristics by a panel of judges for color, taste, aroma, texture, mouth feel, appearance and overall acceptability by following the procedure given by [25]. Bars of different treatments were subjected for sensory evaluation immediately after development. The organoleptic characteristics of bars were determined using a panel of 10 judges of University of Agriculture Faisalabad who were familiar with the major sensory attributes of food products. The panelists were asked to evaluate the products according to parameter mentioned in sensory Performa. The ratings were done on 9 point hedonic scale. 1 = Extremely dislike, 2 = Dislike moderately, 3 = Dislike very much, 4 = Dislike slightly, 5 = Neither like nor dislike, 6 = Like Slightly, 7 = Like moderately, 8 = Like very much and 9 = like extremely. The scores were determined by the judges for each parameter. In the end average score was taken.

## Statistical analysis

The results of the data collected were subjected to variance of analysis under completely randomized design (CRD) for evaluation of means square and mean analytical values and tuckey test for comparison as described by [26].

## Results and discussions

The present study is conducted to develop functional bars by using different ratios of pomegranate peel powder. For this purpose, pomegranate peel powder is analyzed for the proximate composition, selected minerals and TPC, TFC and DPPH. After that bars are prepared and then assessed for proximate composition, mineral, TPC, TFC and DPPH. Proximate analysis for moisture, ash, crude fat, crude fiber and crude protein is also done. For the preparation of bars, pomegranate peel is used in different concentration in the combination of oats. Bars are also subjected to sensory attributes including appearance, color, taste, flavor, texture and overall acceptability to determine the best treatment. The obtained data is then analyzed statistically.

## Proximate analysis of pomegranate peel powder

Mean for proximate analysis of pomegranate peel powder is shown in Table 2. Proximate parameters like moisture, protein, fats, fiber, ash and nitrogen free extract has significant effect on pomegranate peel powder. Pomegranate peel powder has moisture 4.6%, ash 5.31%, crude fat 2.4%, crude protein 6.73%, crude fiber 21% and nitrogen free extract 53.26%. The findings are supported by study [27] reported that pomegranate peel contains moisture 04%, ash 05%, fat 2.4%, crude fiber 21% and protein 9.718%. The higher ash content represent the presence of high mineral contents. Results are in accordance to the study [28] described that pomegranate peel powder contained moisture 9.85%, crude fat 3.27%, crude protein 6.52% and ash 3.53%. In another study [29] the ash content was found to be 5.6%, while the protein and fiber

**Table 2. Proximate composition (%) of pomegranate peel powder.**

| Proximate composition | Pomegranate peel powder |
|---|---|
| Moisture | 4.6±0.22 |
| Ash | 5.31±0.33 |
| Crude fat | 2.4±1.73 |
| Crude protein | 6.73±0.45 |
| Crude fiber | 21±1.03 |
| NFE | 53.26±0.49 |

Mean ± Standard deviation

contents was reported 7.8% and 19% respectively. The higher ash content dictates the high mineral content.

## Total phenolic, total flavonoid and anti-oxidant value of pomegranate peel powder

Mean values of TPC, TFC and DPPH are represented in Table 3 illustrated that pomegranate peel have significant effect. Pomegranate peel contains TPC 348.53 mg GAE/100 g, TFC 269.12 mg QE/100 g and DPPH 37.63%. These results are supported by Salem *et al.* (2020) who reported that TPC and TFC in pomegranate peel powder was 386.47 mg GAE/100 g and 252.26 mg QE/100 g. The results of current study are also supported by research which declared that 352.60 mg GAE/100 g TPC and 304.21 mg QE/100 g TFC were present in pomegranate peel powder [30]. In another study different fruit parts i.e. peel, seeds and whole fruit powder [29]. It was stated that extracts of peel, seeds and whole fruit powder had revealed the presence of alkaloids, glycosides, saponins, carbohydrates, proteins, phenolic compounds and flavonoids. Free amino acids are found to be absent in all the parts of pomegranate. The relevant study revealed that pomegranate peel powder exhibits DPPH activity of 36.61% comparable to our study [31]. Low-temperature air drying caused minimal structural damage, allowing for the retention of more phenolic compounds [32].

## Mineral analysis of pomegranate peel powder

Mean values of minerals i.e. calcium, iron, potassium, magnesium, sodium and zinc are presented in Table 4. Pomegranate peel powder are good source of minerals (Ca. Fe, K, Mg, Na and Zn). Pomegranate peel powder contains calcium 336.84, magnesium 53.2, iron 6.32, potassium 135.5 and phosphorus 112.73 mg/100 g. A study, analyzed the iron content of pomegranate peel powder 9.76 mg/100 g [33]. The results of the study are close to the findings of relevant study in which nutritional and chemical evaluation of dried pomegranate peel was analyzed [28]. Mineral content was analyzed and found that calcium, phosphorus, potassium, magnesium and iron was 342, 120, 150, 56 and 6.11 mg/100 g respectively. The results were

**Table 3. Phytochemical composition of pomegranate peel powder.**

| Phytochemical composition | Pomegranate peel powder |
|---|---|
| TPC | 348.53±0.95 mg GAE/100 g |
| TFC | 269.12±0.73 mg QE/100 g |
| DPPH | 37.63±0.85% |

Mean ± Standard deviation

**Table 4. Mineral composition (mg/100 g) of pomegranate peel powder.**

| Mineral composition | Pomegranate peel powder |
|---|---|
| Calcium | 336.84±0.31 |
| Magnesium | 53.2±1.29 |
| Iron | 6.32± 0.35 |
| Potassium | 135.5±1.03 |
| Phosphorus | 112.73±0.45 |

Mean ± Standard deviation

also supported by relevant research in which mineral content was observed and found that Calcium 338.8, Potassium 143.66 and Iron 5.64 mg/100 g were present [34].

The differences could also be linked to the processing condition, composition and variety of pomegranate peel. Another study in which the pomegranate peel powder was used for the development of cookies [27]. The authors evaluated the pomegranate peel powder for the mineral content and found the potassium 1100 ppm, iron 60 ppm, and zinc 4 ppm. A comparative analysis was performed between seed, peel and fruit powder for mineral content. It was found that phosphorus and zinc had maximum value in peel while sodium, potassium and calcium was abundant in fruit powder [29].

## Proximate analysis of pomegranate peel enriched bars

Mean squares for functional bars proximate *i.e.* moisture, ash, protein, fat, fiber and NFE made from oats and pomegranate peel powder discovered highly significant.

**Moisture.**   Moisture is a good source of water and it is predicted that between 20–30% of total water consumption must come from food. Thus, high moisture levels in foods cause deterioration, so they are susceptible to microbes. The results moisture content showed the increase in moisture content by the addition of pomegranate peel in the bars but the further addition showed decreasing trend as the concentration of pomegranate is increased. These results may be due to the ability of pomegranate peel powder to bind more amount of water. The study correlates the research work in which moisture content of pomegranate peel powder based fiber enriched cupcakes was analyzed [35]. The moisture content of cupcakes was decreased from 13.25% to 11.25%. The research showed that moisture content was decreased as the concentration of pomegranate peel was increased. The study correlates the research work in which moisture content of pomegranate peel powder based sponge cake was analyzed [36]. Hence, research showed that moisture content was decrease due to supplementation of pomegranate peel powder which is low in moisture content, which helps to increase the shelf life of bars. The moisture content of flour is crucial because a small amount of moisture hinders respiration and microbiological activity. Moisture concentrations more than 14% stimulate fungal growth. Lipolytic activity occurs when moisture levels are high, resulting in the loss of nutrients like fat [34].

**Protein content.**   The data showed decline in the protein contents of bars supplemented with pomegranate peel powder. This may due to the presence of low protein content in the pomegranate peel powder and slightly high amount of protein in oats so when we replace oats with peel powder protein content decrease [34]. The maximum value of protein is noted for treatment $T_0$ (9.56%) and minimum value is observed in treatment $T_4$ (8.32%). The protein content in $T_1$, $T_2$ and $T_3$ is 9.42, 9.29 and 8.82% respectively. The results for various treatments are in accordance with the findings of study [37] which observed the decline in protein

percentage from 12.43% to 11.48% in spaghetti with increase in concentration of pomegranate peel powder.

The results of study are close to the findings of relevant study in which effect of pomegranate (*Punica granatum* L.) peel powder meal dietary supplementation on anti-oxidant status and quality of breast meat in broilers was checked [38]. It was concluded by this study that pomegranate peel powder is devoid of fat and protein content. So by its supplementation, the protein content decreases from 23.24% to 20.05%.

**Fat content.** The mean square for fat content of bars enriched with pomegranate peel powder demonstrated a significant relationship among the various treatments utilized to create pomegranate peel enriched bars. Maximum value of fat is noted for treatment $T_0$ (9.05%). The fat content for $T_1$, $T_2$, $T_3$ and $T_4$ are 8.96, 8.91, 8.57 and 7.93% respectively. While minimum is recorded in treatment $T_4$ (7.93%) among different treatments. The data showed that the fat content is decreased with the increase in addition of pomegranate peel powder because pomegranate peel is devoid of fat than fat content in oats.).

The results are closely related to the findings of relevant study in which biscuits were prepared by using pomegranate peel powder to analyze its texture, organoleptic and nutritional properties, reported that control group with 0% pomegranate peel has 16.49% fats and biscuits with 7.5% pomegranate peel powder has 13.48% fats [39]. Identical studies for fat content that the fat content of cookies incorporated with banana peel powder was higher than control cookies [10]. The results of study were close to the findings of supporting study in which effect of pomegranate peel powder supplementation on cookies properties was checked and found that fat content was decreased from 23.78% to 21.33% [40]. It was concluded by this study that pomegranate peel powder is devoid of fat and protein content. So, by its supplementation, the protein and fat content decreases.

**Fiber content.** Dietary fibers can be found in fruits and vegetable wastes. Fibers derived from vegetable and fruits have high amount of soluble dietary fibers. A number of studies showed that utilization of fibers helps to reduce or prevent obesity, some kinds of cancer or tumor and cardiovascular diseases. So, it is necessary to consume 20–35 g fibers daily [35]. The maximum value of fiber is observed in $T_4$ (16.89%). The fiber content for $T_1$, $T_2$, $T_3$ and $T_4$ are 10.490, 11.320, 14.130 and 16.890% respectively. While minimum is noted in $T_0$ (5.23%) treatment which is without any pomegranate peel powder. Adding powder at different levels result in increased the ash and fiber contents. Results revealed that with the increase in the pomegranate peel powder the fiber contents of bars increases. This is because of pomegranate peel as it is abundant reserve of dietary fibers.

Findings of the results are supported by research in which functional snake bars were prepared by using banana peel powder, reported that the fiber content was significantly increased in functional snake bars 9.06% as compared to control bar 5.12% [10]. Similar results were found in research which reported significant increase at the level of (P≤0.05) in the percentage of fiber when the concentrations of pomegranate peels added to the product increased, as the percentage of fiber reached (0.80, 0.95 and 1.18%) for the samples [41]. ($T_1$, $T_2$, and $T_3$) respectively, while the percentage of fiber was 0.54% in the control group ($T_0$). The results of study were close to the relevant study) in which effect of pomegranate peel powder supplementation on cookies properties was checked and found that fiber content was increased from 0.32% to 1.96% [40].

**Ash content.** Mean square of ash content of bars enriches with pomegranate peel powder has been found significant (P<0.05). Mean values for ash contents are ranged from 2.05% to 2.63% presented in Table 4. The data showed increasing trend with the increase of pomegranate peel powder. The highest value is recorded for $T_4$ (2.92%). The ash content for $T_1$, $T_2$, $T_3$

and $T_4$ are 2.63, 2.68, 2.92 and 2.74% while lowest is recorded for $T_0$ (2.05%). Increase in ash content is due to high mineral content inn pomegranate peel powder.

A recent study, support this research and reported that low ash content was present in control cake (0.35%) while highest level of ash content was present in $T_5$ (0.79%) in which 7.5% pomegranate peel powder was supplemented [42]. Study also evaluated that ash content was increase in cupcakes when supplemented with pumpkin seeds [43]. A slight difference may be due to environmental factors and different processing conditions. The results for various treatments are in accordance with the findings of study that observed the decline in protein percentage from 1.32% to 1.56% in spaghetti with increase in concentration of pomegranate peel powder [37].

**Nitrogen free extract.** Mean square of ash content of bars enriches with pomegranate peel powder showed highly significant results for the NFE of bars. The mean values for NFE contents of bars are ranged from 52.62% to 62.51% as shown in Table 5. The maximum value of NFE observed in $T_0$ (64.57%). The values of nitrogen free extract in $T_1$, $T_2$, $T_3$ and $T_4$ are 55.11, 54.53, 53.42 and 52.62% respectively. Minimum NFE is observed in $T_4$ (52.64%). The results showed that nitrogen free extract decrease by increasing pomegranate peel in the bars.

The results are according to the findings of research in which cupcakes were prepared using pomegranate peel powder and observed that the value of NFE of the cakes were decreasing from 58.16 to 54.06% with the increase in concentration of pomegranate peel powder [42]. Results are in accordance to the findings of other research reported that decrease the carbohydrate content in biscuit when supplemented with pomegranate peel powder [44]. The results of study were close to the findings of researches in which effect of pomegranate peel powder supplementation on cookies properties was checked and found that carbohydrate content was decreased from 63.26% to 61.03% [40].

## Phytochemical analysis pomegranate peel enriched bars

**Total phenolic content.** The phenolic compounds have oxidation reduction properties that support the defense mechanism's goal of preventing oxidative damage by suppressing the action of reactive oxygen species which enable them to function as anti-oxidants. Mean square of TPC of developed bars showed highly significant increase in TPC of bars. Pomegranate peel rich in polyphenols which exhibit various biological activities such as anti-fungal, anti-bacterial, anti-viral and anti-inflammatory, anti-allergic, vasodilator and cardio-protective activities [45]. The result of the study showed that the value for TPC by the addition of pomegranate peel are ranged from 142.74–211.79 mg GAE/100 g. The value of TPC for $T_0$, $T_1$, $T_2$, $T_3$ and $T_4$ are 142.74, 159.86, 176.98, 194.63 and 211.79 mg GAE/100 g respectively Results revealed that treatment $T_0$ showed the lowest TPC value (142.74 mg GAE/100 g) whereas treatment $T_4$ has the highest TPC value (211.79 mg GAE/100 g). It is clear from the result that by increasing the amount of pomegranate peel there is increases in the activity of TPC. The study revealed that

**Table 5. Proximate analysis of bars enriched with pomegranate peel powder.**

| Treatment | Moisture content | Protein | Fat content | Fiber content | Ash content | Nitrogen free extract |
|---|---|---|---|---|---|---|
| $T_0$ | 11.60±1.23[c] | 9.56±0.92[a] | 9.05±1.21[a] | 5.23±0.82[d] | 2.05±0.77[a] | 62.51±0.85[a] |
| $T_1$ | 13.38±0.92[a] | 9.42±1.12[a] | 8.96±1.07[a] | 10.49±1.16[c] | 2.63±0.51[d] | 55.11±0.54[b] |
| $T_2$ | 12.33±0.87[b] | 9.29±0.44[a] | 8.91±1.10[a] | 11.32±0.72[c] | 2.68±0.93[d] | 54.53±0.71[bc] |
| $T_3$ | 12.27±1.15[b] | 8.82±1.23[b] | 8.57±0.83[b] | 14.13±1.03[b] | 2.74±1.17[c] | 53.42±1.01[bc] |
| $T_4$ | 11.32±1.03[c] | 8.32±1.14[c] | 7.93±1.08[c] | 16.89±0.64[a] | 2.92±1.25[b] | 52.62±0.93[c] |

pomegranate peel enrichment in biscuits increase the TPC from 36.52 to 71.78 mg GAE/100 g in control group and 10% pomegranate peel powder respectively [46].

**Total flavonoid content.** Mean square for TFC showed a highly significant effect on the TFC of bars among all the treatments. Results are in the range of 129.16–178.23 mg QC/g respectively. For total flavonoid, highest value is recorded in $T_4$ (192.46 mg QE/100 g) in which 20% of pomegranate peel powder is added. The TFC in different treatments $T_1$, $T_2$ and $T_3$ are 143.09, 154.81 and 178.23 mg QE/100 g respectively. While lowest is recorded in $T_0$ (129.16 mg QE/100 g) in which no pomegranate peel powder is added. It is clear from the result that the by increasing the amount of pomegranate peel there is increases in the activity of TFC. Another study revealed that pomegranate peel enrichment in biscuits increase the TFC from 28.45 to 51.13 mg QC/100 g in control group and 10% pomegranate peel powder respectively [46, 47].

**DPPH.** It is a process of determining the activity of anti-oxidant. Anti-oxidant presents in the food play an important role in human body by protecting form harmful free radicals. Pomegranate peel have naturally antioxidant capacity. Ellagic acid, which is abundant in pomegranate peel, is one of the most notable natural antioxidants [48]. Mean square for DPPH showed a highly significant effect on the DPPH of developed bars among all the treatments. The result of the study showed that the value for DPPH by the addition of pomegranate peel in different treatments ($T_0$, $T_1$, $T_2$, $T_3$, $T_4$) are ranged 41.35, 46.58, 52.37, 60.84 and 64.57% respectively as shown in Table 5. Result revealed that treatment $T_0$ showed the lowest DPPH value (41.35%) and treatment $T_4$ has the highest DPPH value (64.57%). It is clear from the result that the by increasing the amount of pomegranate peel there is increases in the activity of DPPH. Study showed that the DPPH radical scavenging amount of spaghetti increased significantly with pomegranate peel powder addition [37]. The radical scavenging capabilities of extract on DPPH are an important indicator of anti-oxidant activity. Another study verified the chemical safety of wheat bread developed using pomegranate peel powder employing the DPPH (Table 6). The results depicted that anti-oxidant capacity of the product ranged between 1.8–6.8 µ mol TEAC/g samples [49]. The antioxidant power also involved in reducing the oxidative stress. Antioxidants play an important role in defense mechanisms related to micronutrients and enzymes systems [50, 51].

## Mineral analysis of bars enriched with pomegranate peel powder

Fruit quality features can be measured in part by their mineral element concentration because minerals are closely related with fruit size, pulp hardness, and soluble solids. Minerals are responsible for good flavor and quality in fruits. Minerals are essential for normal growth and development of human body. Deficiency of minerals causes hidden hunger. Minerals are also essential for cell metabolism, biosynthesis and immune function of human body. For example, iron is major component of myoglobin and hemoglobin. Some minerals cannot be synthesized by human body so must be required by the diet [52]. Fruit peels are rich source of minerals as

**Table 6. Phytochemical analysis pomegranate peel enriched bars.**

| Treatment | TPC | TFC | DPPH |
|---|---|---|---|
| $T_0$ | 142.74±0.65[e] | 129.16±0.64[e] | 41.35±0.82[d] |
| $T_1$ | 159.86±0.76[d] | 143.09±0.76[d] | 46.58±0.77[c] |
| $T_2$ | 176.98±0.91[c] | 154.81±0.93[c] | 52.37±0.91[b] |
| $T_3$ | 194.63±0.86[b] | 178.23±0.86[b] | 60.84±0.72[e] |
| $T_4$ | 211.79±0.63[a] | 192.46±0.53[a] | 64.57±0.69[a] |

compared to the pulp [53]. Variation in minerals level in all treatments of developed product is due to difference in the variety of oats and pomegranate peel powder *i.e.* $T_0$, $T_1$, $T_2$, $T_3$ and $T_4$ contained flour 0%, 5%, 10%, 15%, 20% pomegranate peel powder respectively.

**Calcium.** Calcium is a crucial part of teeth and bones. Calcium is necessary for muscle contraction and relaxation of muscles, healthy neuron functions and immune system functions. Depending on the reference guidelines, dietary reference values for 19 years of age vary from 1000 mg to 1300 mg [54]. Results of study revealed the highly significant relationship of different treatments used for the preparation of bars enriched with pomegranate peel powder. Table 7 demonstrated the average values of calcium ranged from 25.42–31.06 mg/100 g.

Results showed that $T_0$ has lowest value of calcium 25.42 mg/100 g and $T_4$ has highest calcium content 31.06 mg/100 g. The calcium content in different treatments $T_1$, $T_2$ and $T_3$ was 26.81, 28.22, and 29.64 mg/100 g respectively. The increase of calcium content is considered to be related to the high level of calcium present in pomegranate peel powder. Results correlated with findings of study that observed increase in calcium content with the increasing concentration of banana peel powder in formulated snack bar, reported that the calcium content in the formulated snack bar and control bar was 226.89 mg/100 g and 172.58 mg/100 g respectively [10]. Another study highlighted the quality characteristics of biscuits fortified with pomegranate peel was also supported the current results. Study reported that calcium content in biscuits was increased from 62.39 mg/100 g with 0% pomegranate peel to 100.91 mg/100 g with 18% pomegranate peel powder [55, 56].

**Phosphorous.** Study revealed the highly significant relationship of different treatments used for the preparation of bars enriched with pomegranate peel powder. Table 7 illustrated the average values of phosphorus ranged from 51–45.05 mg/100 g. Results showed that $T_0$ has highest value of phosphorus (51.00 mg/100 g) and $T_4$ has lowest phosphorus content (45.05 mg/100 g). The phosphorus content in different treatments $T_1$, $T_2$ and $T_3$ was 49.48, 48.20 and 46.54 mg/100 g respectively. Results indicated decrease in phosphorus content of pomegranate peel enriched bars as the concentration of pomegranate peel powder increases. The decrease of phosphorus content is considered to be related to the low level of phosphorus present in pomegranate peel powder. Current findings agreed with study that investigated the quality characteristics of biscuits fortified with pomegranate peel and found that phosphorus content decreased from 131.13 mg/100 g to 122.24 mg/100 g significantly by increasing the pomegranate peel powder in biscuits formulation. Because pomegranate peel powder is deficit in phosphorus [55, 56].

**Potassium.** The study showed the highly significant relationship of different treatments used for the preparation of pomegranate peel enriched bars. Potassium rich diet helps in the reduction of blood pressure and water retention. Potassium may also aid in the prevention of osteoporosis and kidney stones [57]. Potassium content ranged from 59.46 mg/100 g to 79.15 mg/100 g. Results indicated that $T_0$ 59.46 mg/100 g has lowest potassium content and $T_4$ 79.15 mg/100 g has highest value of potassium. The potassium content in different treatments $T_1$, $T_2$ and $T_3$ is 62.37, 66.45 and 71.32 mg/100 g respectively. These results agreed with work [55, 56]

**Table 7. Mineral analysis of bars enriched with pomegranate peel powder.**

| Treatment | Calcium | Phosphorus | Potassium | Iron | Magnesium |
|-----------|---------|------------|-----------|------|-----------|
| $T_0$ | 25.42±0.63[e] | 51.00±1.01[a] | 59.46±1.13[e] | 1.32±1.20[e] | 54.17±0.88[d] |
| $T_1$ | 26.81±1.42[d] | 49.48±0.61[b] | 62.37±1.11[d] | 1.59±0.96[d] | 54.24±1.21[d] |
| $T_2$ | 28.22±0.84[c] | 48.20±0.79[c] | 66.45±1.22[c] | 1.73±1.23[c] | 55.12±0.79[c] |
| $T_3$ | 29.64±1.26[b] | 46.54±1.31[d] | 71.32±0.31[b] | 1.84±1.22[b] | 56.41±0.71[b] |
| $T_4$ | 31.06±0.58[a] | 45.05±1.09[e] | 79.15±0.28[a] | 1.95±0.83[a] | 57.36±0.68[a] |

which studied the quality characteristics of biscuits fortified with pomegranate peel and they found that potassium content was increased from 125.22 mg/100 g to 269.60 mg/100 g among various treatment in the study.

**Iron.** Iron is a plentiful element on the planet and a biologically important component of all living organisms. For growth and development, the body requires the iron. Iron is an important mineral for the prevention of anemia [58]. Study showed the significant relationship of different treatments used for the preparation of bars enriched with pomegranate peel powder. Table 7 illustrated the average values of iron ranged from 1.32–1.95 mg/100 g. Results showed that $T_0$, $T_1$, $T_2$, $T_3$ and $T_4$ has 1.32, 1.59, 1.73, 1.84 and 1.95 mg/100 g iron content respectively. The results are closely related to the finding of the researchers who studied the quality characteristics of biscuits fortified with pomegranate peel and found that there was significant difference ($p<0.05$) in iron content of the biscuits with pomegranate peel powder addition [55, 56].

**Magnesium.** After calcium, sodium, and potassium, magnesium is the fourth most prevalent mineral in the human body and the second most frequent intracellular cation after potassium. Magnesium is a cofactor in over 300 enzyme systems and is necessary for basic processes such as energy production and nucleic acid synthesis [55]. Results of study revealed the significant relationship of different treatments used for the preparation of bars enriched with pomegranate peel powder. Table 7 illustrated the average values of magnesium ranged from 54.24–57.364 mg/100 g. Results showed that $T_0$, $T_1$, $T_2$, $T_3$ and $T_4$ has 54.17.36, 54.24, 55.12, 56.41 and 57.36 mg/100 g magnesium content respectively. Results of current study are closely related to the findings of study [46] on tomato pomace, mango seeds kernel and pomegranate peels powders for the production of functional biscuits. Study found that magnesium content was increased from 24.44 mg/100 g to 25.03 mg/100 g by adding pomegranate peel powder.

## Texture analysis of bars enriched with pomegranate peel powder

For the acceptability of any product the texture determination is an important factor. The texture of bars was determined by textural analyzer (Mod. TA-XT2 Stable Micro System, Surrey, UK). The bars are bent in order to determine whether any structural change is happened as a result of force exerted on bars. The maximum force is used as an index of hardness. Results of texture analysis showed the significant relationship among the various treatments on hardness. Table 8 showed the mean values for texture analysis of bars ranging from 4.85–7.23. The maximum value for hardness is observed in treatment $T_4$ (7.23). The value of hardness for $T_1$ $T_2$, $T_3$ 5.02, 6.23 and 6.23 respectively. While minimum is recorded in $T_0$ treatment (4.85). This showed that with the increase in the pomegranate peel powder, the hardness level also increases. The reason of increasing hardness of bars is due to high fiber content in pomegranate peel powder [10].

Results are closely related to the findings that the formulation and evaluation of novel functional snack bar with amaranth, rolled oat, and unripened banana peel powder. Results

**Table 8. Texture and color analysis of bars enriched with pomegranate peel powder.**

| Treatment | Texture | L* value | a* value | b* value |
|---|---|---|---|---|
| $T_0$ | 7.43±0.21[d] | 48.93±0.26[a] | 2.93±0.77[d] | 12.49±0.46[a] |
| $T_1$ | 8.32±0.36[c] | 39.64±0.53[b] | 4.16±0.83[cd] | 8.64±1.28[b] |
| $T_2$ | 8.74±0.62[bc] | 37.19±0.75[c] | 5.36±0.64[bc] | 8.13±0.76[b] |
| $T_3$ | 8.96±0.48[b] | 34.46±0.41[d] | 5.92±0.46[b] | 7.69±1.45[c] |
| $T_4$ | 9.17±0.28[a] | 31.06±0.24[e] | 7.88±0.68[a] | 6.24±0.94[d] |

indicated that texture of bar had been effected by adding banana peel powder [10]. It was noted that the hardness of bars was increased with the addition of banana peel. The hardness of control group was 7.456 g, which was increased to 7.715 g in the functional snack bar. Hardness of functional bars was increased due to high fiber present in pomegranate peel powder. These results are in the accordance with the findings of relevant study [59], the hardness was improved with the supplementation of pumpkin flour into bread. In another relevant study, the chewiness of bread was reduced with addition of pumpkin rind and pumpkin seeds [60]. Another study also detected the texture of cookies with the addition of pumpkin seeds and evaluated that it was also improved [61]. Findings of results are supported by a study in which incorporation of soybean flour and pomegranate peel powder by swapping of refined wheat flour improved the soluble as well as insoluble dietary fiber content of cakes [62].

In another study, examination of crude fiber of the snacks expressively increased ($P<0.05$) with elevated fortification level but control sample (100%) had lower value of fiber because of low fiber content in maida (white flour). The maximum fiber content was verified in $T_5$. Snacks contained high fiber content which is needed because fiber is noble source to control digestive problems, cholesterol and lower the risks of cancer [63]. A recent study found that the increase in hardness as the level of the pomegranate peel powder increased in the cupcakes formulation [64]. In another study hardness and springiness of the capsicum powder supplemented muffin samples was investigated and reported decreasing trend in hardness and springiness value (0.82 to 0.76), Volume of crumb and air cells in the muffins are two factors that affect the hardness of the muffins according to [65, 66]

## Color analysis of bars enriched with pomegranate peel powder

**L\* value.** Color is one of the important factor as it is use to check the quality of food. The color of food also play role in the visual acceptability of food. L* is the lightness value (dark (0) to white (100)). Mean square of L* value for bars enriched with peel powder reported that there is highly significant difference between control and treatments groups. Mean values of L* are shown in Table 8. Treatment $T_0$ (48.93) showed the highest L* value and treatment $T_4$ (34.46) has the lowest L* value. The color L*of bars measured for $T_0$, $T_1$, $T_2$, $T_3$ and $T_4$ are 48.93, 39.64, 37.19, 34.46 and 31.06 respectively. It is evident that with the increases of pomegranate peel there is decrease in the value of L*. The changes in L * value in PPP-fortified snacks may be due to changes in anthocyanidin content in pomegranate peels or Millard reactions that occurred when baking. The result of study is correlated with the finding of who prepared cupcakes by adding pomegranate peel powder in the ratio of 5, 10, 15 and 20%. The results showed that the pomegranate peel powder had positive effect on the value of color L* [35].

**a\* value.** a* signifies greenness/redness. Mean square of a* value for bars enriched with peel powder reported that there is highly significant difference between control and treatments groups. Mean square of a* value for bars enriched with peel powder is represented in Table 8. Treatment $T_0$ (2.93) showed the lowest a* value and treatment $T_4$ (7.88) has the highest a* value. Findings for a* values in different treatments ranged from 7.52 to 11.45. a* value of bars increased in order $T_0$, $T_1$, $T_2$, $T_3$ and $T_4$ are 2.93, 4.16, 5.36, 5.92 and 7.88 respectively. a* value showed the intensity of redness color. It is evident that with the increases of pomegranate peel there is increase in the value of a*. The incorporation of PPP increased the redness of biscuits, as predicted due to its anthocyanidin concentration. The researchers observed a change in the crust color. According to their findings, Maillard reactions during baking may disguise the redness [56]. The result of study is correlated with the finding of who prepared cupcakes by adding pomegranate peel powder in the ratio of 5, 10, 15 and 20%. The results showed that the pomegranate peel powder have positive effect on the value of color a* [35].

**b\* value.** b\* signifies yellowness/blueness. Mean square of b\* value for bars enriched with peel powder reported that there is significant difference between control and treatments groups. Mean square of b\* value for bars enriched with peel powder is represented in Table 8. Treatment $T_0$ showed the highest b\* value 12.49 and treatment $T_4$ has the lowest b\* value 6.24. Findings for b\* values in different treatments ranged from 7.52 to 11.45. The color b\*of bars measured for different treatments $T_0$, $T_1$, $T_2$, $T_3$ and $T_4$ are 12.49, 8.64, 8.13, 7.69 and 6.24 respectively. It is evident that with the increase of pomegranate peel there is decrease in the value of b\*. Decreasing trend was observed in the value of b\* with the addition of pomegranate peel powder. Anthocyanidins change color with pH and temperature. It was assumed that variations in b values coincided with changes in anthocyanidin levels [56]. The result of study is correlated with the findings of Gadallah *et al.* (2022) who prepared cupcakes by adding pomegranate peel powder in the ratio of 5, 10, 15 and 20%. The results showed that the pomegranate peel powder have positive effect on the value of color b\* [35].

## Sensory analysis of bars enriched with pomegranate peel powder

**Color.** Color of any food product is most important because at first color is seen and then food is eaten. Color is a very important property defining the consumer acceptability [33]. Mean square for color of bars enriched with peel powder is represented that there is significant difference between control and treatments groups. Mean values for color of bars have been presented in Table 9. The values are ranged from 6.95 to 8.86 for different treatments. It showed from the table that with the increase in the levels of pomegranate peel powder, the acceptance of color significantly decreases. Treatment $T_3$ (8.86) obtained the maximum score while $T_4$ (6.95) got the minimum score. The color score for $T_0$, $T_1$, $T_2$, $T_3$ and $T_4$ are 7.3, 7.65, 8.81, 8.86 and 6.65 respectively. As $T_0$ is in whitish color as no addition of pomegranate peel powder decrease the acceptability whereas $T_2$ (8.81) and $T_3$ (8.86) showed highest acceptability due to color with addition of pomegranate peel powder but the trend declined as the color darkens towards $T_4$ which is not acceptable to the panels. It is important to note that color is a crucial attribute in determining consumer acceptance and preference for food products. Previous research has shown that the incorporation of natural ingredients in bread formulations can improve appearance/color attributes. For instance, a study investigated the effect of pomegranate peel powder from 1% to 10% on color of biscuits. The color of chapatti was more acceptable that is 8.6% when 5% PPP was added [46].

**Texture.** Texture is an important parameter used in processed and fresh foods in food industry to check the quality and acceptability of product. Mean square for texture of bars enriched with peel powder reported that there is significant difference between control and treatments groups. The mean values of the current research described in Table 9. Among the texture characteristics, hardness (firmness) is one of the most important parameters of fruit and vegetables, which is often used to check the freshness of food. It demonstrated the range of texture from 6.1 to 8.5. It imply that the treatment $T_3$ showed highest 8.5 score and lowest for

**Table 9. Sensory analysis of bars enriched with pomegranate peel powder.**

| Treatment | Color | Texture | Flavor |
|---|---|---|---|
| $T_0$ | 7.30±1.01[ab] | 7.1±1.02[bc] | 6.7±1.02[b] |
| $T_1$ | 7.65±0.72[ab] | 7.6±0.48[ab] | 7.4±0.60[ab] |
| $T_2$ | 8.81±1.23[a] | 7.9±0.57[ab] | 7.5±1.25[ab] |
| $T_3$ | 8.86±1.17[a] | 8.5±1.08[a] | 8.8±0.12[a] |
| $T_4$ | 6.65±1.43[b] | 6.1±0.63[c] | 6.2±1.13[b] |

$T_4$ 6.1. Higher texture scores compared to other treatments are most likely the outcome of the pomegranate peel powder addition, which likely helped create a preferable texture. Texture is observed for $T_0$, $T_1$, $T_2$, $T_3$ and $T_4$ as 7.1, 7.6, 7.9, 8.5 and 6.1 respectively. A study found an association between textural stiffness and fiber content in processed cake products Gadallah *et al.* (2022) [35].

The whole eating experience and the acceptance of food products by consumers are significantly influenced by texture, an essential sensory quality. Previous studies have shown how different components affect the texture of bread products. A study showed similar results. They conducted a research on oats bars and added banana peel powder from 5 to 25 and texture was observed. Best texture of 8.6 was observed by addition of 15 banana peel powder [10].

**Flavor.** Flavor is most important in choice of food to be eaten. The mean square for flavor of the current research showed highly significant results. Mean values for flavor are shown in Table 9. It implies that the treatment $T_3$ showed highest 8.80 score and lowest for $T_4$ 6.20. Flavor score is observed as 6.70, 7.40, 7.50, 8.80, and 6.20 for $T_0$, $T_1$, $T_2$, $T_3$ and $T_4$ respectively. The treatments have a significant influence on the flavor scores. The findings show that the inclusion of pomegranate peel powder in treatment $T_3$ considerably improved the flavor of the pomegranate peel. Compared to other treatments, the pomegranate peel powder addition most certainly enhanced the taste profile and raised the flavor scores. But too much peel powder can affect the flavor because PPP's phenolic content caused an acidic flavor in biscuits, affecting taste ratings [56]. A key sensory quality that affects customer acceptance and preference for food items is flavor. Previous studies have shown how natural substances may improve the flavor of biscuits. A study showed the effect of pomegranate peel powder on biscuits. Flavor of biscuits was acceptable that is 8.5 score when 5% pomegranate peel was added [46].

## Overall acceptability

The overall acceptability of the bars enriched with pomegranate peel powder indicated highly significant results. Mean square for overall acceptability of bars enriched with peel powder reported that there is significant difference between control and treatments groups. These results indicated in Table 10 showed that the treatment $T_3$ has highest 8.90 and $T_0$ has lowest 6.00. The result for all other treatments as 6.00, 6.20, 6.70, 8.90 and 7.50% for $T_0$, $T_1$, $T_2$, $T_3$ and $T_4$ respectively. $T_3$ received the highest overall acceptability scores, suggesting that it is preferred by the sensory panel compared to other treatments. As described in a study Panelists reported minor bitterness and sourness in PP18 biscuits [56]. Which suggests that high citric acid and low pH levels might promote sourness in meals. So it may affect the overall acceptability of $T_4$. Overall acceptability is a crucial attribute in determining consumer preference and willingness to consume a food product. A recent study showed similar results. They conducted a research on oats bars and added banana peel powder from 5% to 25% and overall acceptability was observed. Best texture of 8.5 was observed by addition of 15% banana peel powder [10].

**Table 10. Overall acceptability of bars enriched with pomegranate peel powder.**

| Treatment | Overall acceptability |
|---|---|
| $T_0$ | 6.0±1.08[b] |
| $T_1$ | 6.2±1.56[b] |
| $T_2$ | 6.7±0.56[b] |
| $T_3$ | 8.9±0.84[a] |
| $T_4$ | 7.5±1.33[ab] |

## Conclusion

The study successfully developed snack bars incorporating pomegranate peel powder and oats, which offer substantial health benefits. These bars exhibit enhanced nutritional profiles, particularly with increased fiber, TPC, TFC, and DPPH. The incorporation of pomegranate peel powder resulted in significant improvements in mineral content, particularly calcium, potassium, and magnesium, while slightly reducing protein and fat content. Among the five formulations tested, the bar containing 15% pomegranate peel powder (T3) emerged as the most preferred based on sensory evaluation, indicating high overall acceptability. This formulation provides a rich source of phytochemicals, minerals, and fibers, aligning with the growing consumer demand for functional foods that contribute to overall health and well-being. At $T_3$ composition of bars nutritional benefits of PPP are maximized without negatively impacting the taste, texture, or appearance. Too much PPP might introduce a bitter taste or gritty texture, while too little may not provide enough functional benefits. The study highlights the potential of pomegranate peel as a valuable ingredient in enhancing the nutritional quality of cereal-based snack bars.

## Supporting information

**S1 Data.**
(XLSX)

**S1 File. Inclusivity in global research.**
(DOCX)

## Acknowledgments

Authors are thankful University of Agriculture Faisalabad, Pakistan for providing the research facilities for this study. We confirm the final authorship for this manuscript, and we ensure that anyone else who contributed to the manuscript but does not qualify for authorship has been acknowledged with their permission. We acknowledge that all listed authors have made a significant scientific contribution to the research in the manuscript approved its claims and agreed to be an author.

## Author Contributions

**Conceptualization:** Farhan Saeed, Amar Shankar, Rutaba Nadeem, Ashish Singh Chauhan, Ali Imran, Abdela Befa Kinki.

**Data curation:** Muhammad Afzaal.

**Formal analysis:** Jaspreet Kaur.

**Supervision:** Muhammad Aamir.

**Writing – original draft:** Rameeza Abbas.

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
