## [Decision Letter · Decision Letter 0]

29 Oct 2024

PONE-D-24-32412DEVELOPMENT AND NUTRITIONAL EVALUATION OF POMEGRANATE PEEL ENRICHED BARSPLOS ONE

Dear Dr. Kinki,

Thank you for submitting your manuscript to PLOS ONE. After careful consideration, we feel that it has merit but does not fully meet PLOS ONE’s publication criteria as it currently stands. Therefore, we invite you to submit a revised version of the manuscript that addresses the points raised during the review process.

We look forward to receiving your revised manuscript.

Kind regards,

Lakshmanan Govindan

Academic Editor

PLOS ONE

Journal requirements: When submitting your revision, we need you to address these additional requirements. 1. Please ensure that your manuscript meets PLOS ONE's style requirements, including those for file naming. The PLOS ONE style templates can be found at https://journals.plos.org/plosone/s/file?id=wjVg/PLOSOne_formatting_sample_main_body.pdf and https://journals.plos.org/plosone/s/file?id=ba62/PLOSOne_formatting_sample_title_authors_affiliations.pdf 2. Please include a complete copy of PLOS’ questionnaire on inclusivity in global research in your revised manuscript. Our policy for research in this area aims to improve transparency in the reporting of research performed outside of researchers’ own country or community. The policy applies to researchers who have travelled to a different country to conduct research, research with Indigenous populations or their lands, and research on cultural artefacts. The questionnaire can also be requested at the journal’s discretion for any other submissions, even if these conditions are not met.  Please find more information on the policy and a link to download a blank copy of the questionnaire here: https://journals.plos.org/plosone/s/best-practices-in-research-reporting. Please upload a completed version of your questionnaire as Supporting Information when you resubmit your manuscript. 3. Thank you for stating the following in your Competing Interests section:  [no]. Please complete your Competing Interests on the online submission form to state any Competing Interests. If you have no competing interests, please state ""The authors have declared that no competing interests exist."", as detailed online in our guide for authors at http://journals.plos.org/plosone/s/submit-now  This information should be included in your cover letter; we will change the online submission form on your behalf. 4. Please provide a complete Data Availability Statement in the submission form, ensuring you include all necessary access information or a reason for why you are unable to make your data freely accessible. If your research concerns only data provided within your submission, please write "All data are in the manuscript and/or supporting information files" as your Data Availability Statement.

Additional Editor Comments:

Major Revision

Reviewers' comments:

Reviewer's Responses to Questions

**Comments to the Author**

1. Is the manuscript technically sound, and do the data support the conclusions?

Reviewer #1: Partly

Reviewer #2: No

Reviewer #3: Yes

Reviewer #4: Partly

Reviewer #5: Yes

Reviewer #6: Partly

2. Has the statistical analysis been performed appropriately and rigorously? 

Reviewer #1: No

Reviewer #2: No

Reviewer #3: Yes

Reviewer #4: I Don't Know

Reviewer #5: Yes

Reviewer #6: Yes

3. Have the authors made all data underlying the findings in their manuscript fully available?

Reviewer #1: No

Reviewer #2: No

Reviewer #3: Yes

Reviewer #4: Yes

Reviewer #5: Yes

Reviewer #6: Yes

4. Is the manuscript presented in an intelligible fashion and written in standard English?

Reviewer #1: Yes

Reviewer #2: No

Reviewer #3: No

Reviewer #4: No

Reviewer #5: Yes

Reviewer #6: Yes

5. Review Comments to the Author

Reviewer #1: This study investigated the effect of the addition of pomegranate peel powder on the bars quality. The abstract needs to be rewritten. The materials and methods section is poorly written. The manuscript is written very sloppily. The scientific quality is too low. The discussion is insufficient.

- Line and page numbers should be added.

- The abstract must include the background, methods, some results and conclusions. Please, improve the abstract. Please give more details about the formulations (T0-T4) in the abstract. The abstract does not reflect the all results of the study. It should be improved.

- What is the novelty of the study? Please explain it at the end of the Introduction section.

- In the Materials and Methods section, Please add the brand, model and origin of all equipment.

- The preparation of bars should be detailed.

- The formulations of the bars should be added to the Materials and Methods section.

- Please give more details about the analysis methods of total phenolic, total flavonoid contents and antioxidant activity.

- The materials and methods section should be completely rewritten.

- Abbreviations (TPC, TFC, DPPH) should be defined at first mention and used consistently thereafter.

- Please add the units of the moisture, ash, fat and fiber content in the sentence of “The findings are supported by Ullah et al. (2012) who reported that pomegranate peel contains moisture 04, ash 05, fat 2.4, crude fiber 21 and protein 9.718%.”.

- Antioxidant? or Anti-oxidant? Please select one.

- Please add a space between the unit and the number “348.53 mg GAE/100 g”.

- “The reason of increasing hardness of bars is due to high fiber content in pomegranate peel powder.” Please add a reference.

- The discussion of color results should be improved.

- In Table 7, Please check “8.96±0.48c”. Is it a?

Reviewer #2: Very poorly written and presented work! Low quality of data without novelty, I can't accept it!

Please check the whole manuscript carefully and try to modify it from top to bottom! It would be great if the authors check the language too!

Reviewer #3: overall paper is good and well explained and sufficient experiement with results related to this kind of study. I will suggest a detailed evaluation of english language improvement for better understanding and a language quality for the journal standard.

Reviewer #4: I have reviewed the manuscript, and I find this a very interesting study to repurpose food by-product. A good effort towards reducing food waste. The study successfully developed a formulation that will be beneficial and accepted by the community.

The manuscript, however, lacks clarity and explanation of the observed results could be improved. Methodologies employed in the study are over-simplified and the actual statistical analysis was not defined.

Suggestions for improvement:

1. Page 13 – ‘The moisture content of cupcakes was decreased from 9.84 to 13.25%’. Please clarify this sentence; increased or decreased? If this is an ‘increased’, please re-explain the conclusion for this part i.e. as to how it affects the shelf life of the formulated pomegranate bar

2. Page 13-14 – Protein and fat content. Authors mentioned that the pomegranate peel devoid protein and fat content. This statement does not support the observed reduction pattern of protein and fat content in the pomegranate bar. Increment of the pomegranate peels will also mean more protein and fat will be added to the bar. Biochemical reactions and processes involved in preparation of the bar could be worth investigating as they may provide better explanation to the observed results.

3. Page 18 – DPPH evaluation. Please re-check the DPPH value for T0. There’s a difference between the value stated in the paragraph and table 5.

4. Pages 20 and 21 – Phosphorus and magnesium evaluation. Please re-clarify the reason for the observed reduction of these minerals in the pomegranate bar.

5. Please give a simple description of the color i.e. L*, a* and b* values.

I strongly suggest language (English) editing throughout the manuscript.

Reviewer #5: Pomegranate peel, often considered a waste product, is gaining attention for its bioactive compounds. The study used well-established methods, emphasizing the potential application of pomegranate peel as a functional food ingredient. Therefore, the study is novel and relevant. However, manuscript should be improved:

1- The study mentions sourcing pomegranate peel, jaggery, and oats from local markets. However, there’s no indication of control over the quality and variability of these raw materials, which could affect reproducibility and consistency in the results. Were the pomegranates from a single variety or region?

2- While moisture content is measured for pomegranate peel powder, it would be valuable to also assess the moisture stability of the final product (bars). High moisture content could affect shelf life, microbial growth, and texture.

3- Storage and stability, microbial load and toxicity should be studied.

4- The manuscript lacks experimental design, details, and clarifications of the methods used.

5- The statistical analysis was not clarified.

6- The washing, drying, and grinding processes of the pomegranate peel could impact its bioactive compounds. Did the drying temperature affect the phenolic content? More details on the temperature and conditions used should be incorporated in the manuscript.

7- The abstract lacks specificity in results, vague statistical analysis. The abstract did not specify the evaluated parameters.

8- Repetition in the content of the results and discussion. Discussion lacks in-depth interpretation: How do the higher fibre, phenolic, and antioxidant levels in pomegranate peel powder impact the functional and health aspects of the final product? Why is T3 (15% pomegranate peel) the most acceptable in sensory evaluation? Is there a trade-off between nutritional enhancement and sensory properties (taste or texture)?

9- findings should be integrated with the final enriched bars.

10- More detail on how the addition of pomegranate peel powder influenced the bar composition is needed.

11- more discussion and interpretation should be done on sensory evaluation.

12- The manuscript could benefit from more introductory statements along the texts.

Reviewer #6: Abstract: I don't see the 'why' for this study. There are plenty of nutritional bars that exist, what is the benefit that this nutritional bar will have over others?

Introduction:

1. I would change the opening sentence. It won't be perceived well by certain audiences that are reading this.

2. I would look into costs of foods whether it be from the countries of the authors or from others. Cost in the US (especially pomegranate) are costly and would not necessarily be a solution for food disparities.

3. Are there any clinical trials that can be included as reference for oats? This would gain better emphasis on the use of it.

4. The paragraph starting with "The consumption of snacks" should be taken out as it's not adding it in the placement of where it is at in the introduction. It can be moved to another place but that's for readability.

Methods

1. Specificity of the product should be given as much as possible (for example, are the oats pressed or are they raw?)

2. Please mention the country for Faisalabad

3. Please state what AOAC is. There are no explanation of this acronym before.

4. Was the method mentioned was the one that was followed? Were there any modifications?

5. Is there a way to have a figure of the workflow? It'll help with the explanation that is given in this section.

6. No mentions of any statistical methods used with the analysis.

Results and Discussion

I think this was the best part of the paper and do not see much edits needed in this section.

Conclusion

Not much here as it is short.

Overall, this is a unique publication and would like to see the edits metioned before being accepted.

6. PLOS authors have the option to publish the peer review history of their article (what does this mean?). If published, this will include your full peer review and any attached files.

Reviewer #1: No

Reviewer #2: **Yes: **Dr. Arnab Banerjee

Reviewer #3: No

Reviewer #4: No

Reviewer #5: **Yes: **Amel Elbasyouni

Reviewer #6: **Yes: **Susan Egbert

---

## [Author Response · Author response to Decision Letter 0]

28 Nov 2024

Dear Editor,

Here we are submitting our revised review article entitled “DEVELOPMENT AND NUTRITIONAL EVALUATION OF POMEGRANATE PEEL ENRICHED BARS”. The changes in the manuscript are highlighted with Green, Grey, pink, yellow, blue and red. We have tried our best to improve all the issues highlighted by the reviewers/editors. We will be happy to make any further suggested changes. Looking forward for your kind considerations. 

 Reviewer-1

Comments

1. This study investigated the effect of the addition of pomegranate peel powder on the bars quality. The abstract needs to be rewritten. The materials and methods section is poorly written. The manuscript is written very sloppily. The scientific quality is too low. The discussion is insufficient.

Dear reviewer, thanks for your suggestions. As per your suggestions, abstract and materials re-written.

2. Line and page numbers should be added.

Dear reviewer, thanks for your suggestions. As per your suggestions line number and page number has been added.

3. The abstract must include the background, methods, some results and conclusions. Please, improve the abstract. Please give more details about the formulations (T0-T4) in the abstract. The abstract does not reflect the all results of the study. It should be improved.

Dear reviewer, thanks for your suggestions. As per your suggestions, the abstract section has been improved. Formulation have been explained.

4. What is the novelty of the study? Please explain it at the end of the Introduction section.

Dear reviewer, thanks for your suggestions. As per your suggestions, the novelity of the study have been mentioned

5. In the Materials and Methods section, please add the brand, model and origin of all equipment.

Dear reviewer, thanks for your suggestions, As per your suggestions, the brand, model and origin of all equipment has been added.

6. The preparation of bars should be detailed. The formulations of the bars should be added to the Materials and Methods section.

Dear reviewer, thanks for your suggestions, I have mention composition and preparation procedure of bars. 

7. Please give more details about the analysis methods of total phenolic, total flavonoid contents and antioxidant activity.

Dear reviewer, thanks for your suggestions, I have considered this suggestion.

8. The materials and methods section should be completely rewritten.

Dear reviewer, thanks for your suggestions, I have rewritten the materials and methods section.

9. - Abbreviations (TPC, TFC, DPPH) should be defined at first mention and used consistently thereafter.

Dear reviewer, thanks for your suggestions. As per your suggestions, the TPC, TFC and DPPH (consistently mentioned) has been added.

10. - Please add the units of the moisture, ash, fat and fiber content in the sentence of “The findings are supported by Ullah et al. (2012) who reported that pomegranate peel contains moisture 04, ash 05, fat 2.4, crude fiber 21 and protein 9.718%.”.

Respectable reviewer, thanks for your suggestions. The units have mentioned consistently throughout the article.

11. - Antioxidant? or Anti-oxidant? Please select one.

Respectable reviewer, thanks for your suggestions. I have used Anti-oxidant.

12. - Please add a space between the unit and the number “348.53 mg GAE/100 g”.

Dear reviewer, thanks for your suggestions. Its been done.

13. “The reason of increasing hardness of bars is due to high fiber content in pomegranate peel powder.” Please add a reference.

Dear reviewer, thanks for your suggestions. Its been done.

14. The discussion of color results should be improved.

Dear reviewer, thanks for your suggestions. Its been improved.

15. In Table 7, Please check “8.96±0.48c”. Is it a?

Respectable reviewer, this have been improved.

Reviewer 2

1. Very poorly written and presented work! Low quality of data without novelty, I can't accept it! Please check the whole manuscript carefully and try to modify it from top to bottom! It would be great if the authors check the language too!

Dear reviewer, thanks for your kind suggestions. The manuscript has now been revised novelity have been mentoned and I have modified from top to bottom. And language have been improved

Reviewer-3

1. Overall paper is good and well explained and sufficient experiement with results related to this kind of study. I will suggest a detailed evaluation of english language improvement for better understanding and a language quality for the journal standard.

Dear reviewer, thanks for your kind suggestions. Yes, the language quality have been improved. 

Reviewer 4

1. Page 13 – ‘The moisture content of cupcakes was decreased from 9.84 to 13.25%’. Please clarify this sentence; increased or decreased? If this is an ‘increased’, please re-explain the conclusion for this part i.e. as to how it affects the shelf life of the formulated pomegranate bar

Dear reviewer, thanks for your kind suggestions. Yes, decrease trend and reason have been mentioned properly. 

2. Page 13-14 – Protein and fat content. Authors mentioned that the pomegranate peel devoid protein and fat content. This statement does not support the observed reduction pattern of protein and fat content in the pomegranate bar. Increment of the pomegranate peels will also mean more protein and fat will be added to the bar. Biochemical reactions and processes involved in preparation of the bar could be worth investigating as they may provide better explanation to the observed results.

Dear reviewer, thanks for your kind suggestions. But as I have mentioned that peel powder devoid protein and fat. So, when I have increase its content the oats percentage have decreased which is the reason the protein and fat content have followed a decrease trend.

3. Page 18 – DPPH evaluation. Please re-check the DPPH value for T0. There’s a difference between the value stated in the paragraph and table 5.

Dear reviewer, thanks for your kind suggestions. I have been re-written.

4. Pages 20 and 21 – Phosphorus and magnesium evaluation. Please re-clarify the reason for the observed reduction of these minerals in the pomegranate bar.

Dear reviewer, thanks for your kind suggestions. I have re-evaluated these results. 

Please give a simple description of the color i.e. L*, a* and b* values.

Dear reviewer, thanks for your kind suggestions. I have mentioned

Reviewer 5

1- The study mentions sourcing pomegranate peel, jaggery, and oats from local markets. However, there’s no indication of control over the quality and variability of these raw materials, which could affect reproducibility and consistency in the results. Were the pomegranates from a single variety or region?

Dear reviewer, thanks for your suggestion. I have mention the source which I miss mentioned previously. And also local variety of pomegranate used in food industry have been mentioned.

2- While moisture content is measured for pomegranate peel powder, it would be valuable to also assess the moisture stability of the final product (bars). High moisture content could affect shelf life, microbial growth, and texture.

2- Storage and stability, microbial load and toxicity should be studied.

Dear reviewer, thanks for your kind suggestions. While our current study focuses on the development and characterisation of pomegranate peel powder and its application in bars, we recognise that evaluating these parameters is crucial for a comprehensive understanding of the product's nutritional value. But we plan to investigate these aspects in detail, including moisture stability, microbial load, and potential toxicity, in our future studies.

4- The manuscript lacks experimental design, details, and clarifications of the methods used.

Dear reviewer, thanks for your kind suggestions. I have re-written the material and method part

5- The statistical analysis was not clarified.

Dear reviewer, thanks for your kind suggestions. I have improved it.

6- The washing, drying, and grinding processes of the pomegranate peel could impact its bioactive compounds. Did the drying temperature affect the phenolic content? More details on the temperature and conditions used should be incorporated in the manuscript.

Dear reviewer, thanks for your kind suggestions. I have mentioned it. Pomegranate peel is dried at room temperature under shade and have little impact on its bioactive compounds.

7- The abstract lacks specificity in results, vague statistical analysis. The abstract did not specify the evaluated parameters.

Dear reviewer, thanks for your kind suggestions. I have re-write it.

8- Repetition in the content of the results and discussion. Discussion lacks in-depth interpretation: How do the higher fibre, phenolic, and antioxidant levels in pomegranate peel powder impact the functional and health aspects of the final product? Why is T3 (15% pomegranate peel) the most acceptable in sensory evaluation? Is there a trade-off between nutritional enhancement and sensory properties (taste or texture)?

Dear reviewer, thanks for your kind suggestions. Health benfits of higher fibre, phenolic, and antioxidant levels have been mentioned. At T3 composition of bars nutritional benefits of PPP are maximized without negatively impacting the taste, texture, or appearance. Too much PPP might introduce a bitter taste or gritty texture, while too little may not provide enough functional benefits.

9- findings should be integrated with the final enriched bars.

Thank you for your suggestions. We have tried our best to make it possible. We have reviewed all the results and discussion has been updated.

10- More detail on how the addition of pomegranate peel powder influenced the bar composition is needed.

Thank you for your suggestions. We have tried our best to make it possible. We have reviewed all the results and discussion has been updated.

11- more discussion and interpretation should be done on sensory evaluation.

Thank you for your suggestion. I have improved the discussion.

12- The manuscript could benefit from more introductory statements along the texts.

Thank you for your suggestions. We have tried our best to make it possible. We have reviewed all the results and discussion has been updated.

Reviewer #6

Abstract: I don't see the 'why' for this study. There are plenty of nutritional bars that exist, what is the benefit that this nutritional bar will have over others?

Introduction:

1. I would change the opening sentence. It won't be perceived well by certain audiences that are reading this. Intro 1st line rewritten

Dear reviewer, thanks for your kind suggestions. I have re-write it.

2. I would look into costs of foods whether it be from the countries of the authors or from others. Cost in the US (especially pomegranate) are costly and would not necessarily be a solution for food disparities.

Dear reviewer, thanks for your kind suggestions. As "Pomegranate peel powder (PPP) is a low-cost by-product of the pomegranate juice industry, typically discarded as waste. Utilizing PPP helps reduce costs and supports sustainability by repurposing food waste, making it an economically viable ingredient irrespective of regional differences in the price of whole pomegranates."

3. Are there any clinical trials that can be included as reference for oats? This would gain better emphasis on the use of it.

Dear reviewer, thanks for your kind suggestions. I has been added.

4. The paragraph starting with "The consumption of snacks" should be taken out as it's not adding it in the placement of where it is at in the introduction. It can be moved to another place but that's for readability.

Dear reviewer, thanks for your kind suggestions. I have improved it.

Methods

1. Specificity of the product should be given as much as possible (for example, are the oats pressed or are they raw?)

Dear reviewer, thanks for your kind suggestions. I have mentioned it.

2. Please mention the country for Faisalabad

Dear reviewer, thanks for your kind suggestions. I have mentioned it.

3. Please state what AOAC is. There are no explanation of this acronym before.

Dear reviewer, thanks for your kind suggestions. It’s the reference manual that I have used.

4. Was the method mentioned was the one that was followed? Were there any modifications?

Dear reviewer, thanks for your kind suggestions. I have used the procedures from the manual of AOAC.

5. Is there a way to have a figure of the workflow? It'll help with the explanation that is given in this section.

Dear reviewer, thanks for your kind suggestions. I have mentioned it.

6. No mentions of any statistical methods used with the analysis.

Dear reviewer, thanks for your kind suggestions. I have mentioned it.

Results and Discussion

I think this was the best part of the paper and do not see much edits needed in this section.

Conclusion: Not much here as it is short.

---

## [Editor Report · Decision Letter 1]

2 Dec 2024

DEVELOPMENT AND NUTRITIONAL EVALUATION OF POMEGRANATE PEEL ENRICHED BARS

PONE-D-24-32412R1

Dear Dr. Kinki,

We’re pleased to inform you that your manuscript has been judged scientifically suitable for publication and will be formally accepted for publication once it meets all outstanding technical requirements.

Kind regards,

Lakshmanan Govindan

Academic Editor

PLOS ONE

Additional Editor Comments (optional):

Accept
---

## [Editor Report · Acceptance letter]

13 Dec 2024

PONE-D-24-32412R1 

PLOS ONE

Dear Dr. Kinki, 

I'm pleased to inform you that your manuscript has been deemed suitable for publication in PLOS ONE. Congratulations! Your manuscript is now being handed over to our production team.

Kind regards, 

on behalf of

Dr. Lakshmanan Govindan 

Academic Editor

PLOS ONE